# Memory Replay GANs: learning to generate images from new categories without forgetting

**Chenshen Wu, Luis Herranz, Xialei Liu, Yaxing Wang,**
**Joost van de Weijer, Bogdan Raducanu**
Computer Vision Center
Universitat Autònoma de Barcelona, Spain
{chenshen, lherranz, xialei, yaxing, joost, bogdan}@cvc.uab.es

## Abstract

Previous works on sequential learning address the problem of forgetting in discriminative models. In this paper we consider the case of generative models. In particular, we investigate generative adversarial networks (GANs) in the task of learning new categories in a sequential fashion. We first show that sequential fine tuning renders the network unable to properly generate images from previous categories (i.e. forgetting). Addressing this problem, we propose *Memory Replay GANs* (MeRGANs), a conditional GAN framework that integrates a memory replay generator. We study two methods to prevent forgetting by leveraging these replays, namely *joint training with replay* and *replay alignment*. Qualitative and quantitative experimental results in MNIST, SVHN and LSUN datasets show that our memory replay approach can generate competitive images while significantly mitigating the forgetting of previous categories. [1]

## 1   Introduction

Generative adversarial networks (GANs) [6] are a popular framework for image generation due to their capability to learn a mapping between a low-dimensional latent space and a complex distribution of interest, such as natural images. The approach is based on an adversarial game between a generator that tries to generate good images and a discriminator that tries to discriminate between real training samples and generated. The original framework has been improved with new architectures [21, 9] and more robust losses [2, 7, 16].

GANs can be used to sample images by mapping a randomly sampled latent vector. While providing diversity, there is little control over the semantic properties of what is being generated. Conditional GANs [18] enable the use of semantic conditions as inputs, so the semantic properties and the inherent diversity can be decoupled. The simplest condition is just the category label, allowing to control the category of the generated image [20].

As most machine learning problems, image generation models have been studied in the conventional setting that assumes all training data is available at training time. This assumption can be unrealistic in practice, and modern neural networks face scenarios where tasks and data are not known in advance, requiring to continuously update their models upon the arrival of new data or new tasks. Unfortunately, neural networks suffer from severe degradation when they are updated in a sequential manner without revisiting data from previous tasks (known as *catastrophic forgetting* [17]). Most strategies to prevent forgetting in neural networks rely on regularizing weights [4, 14] or activations [13], keeping a small set of exemplars from previous categories [22, 15], or memory replay mechanisms[23, 25, 10].

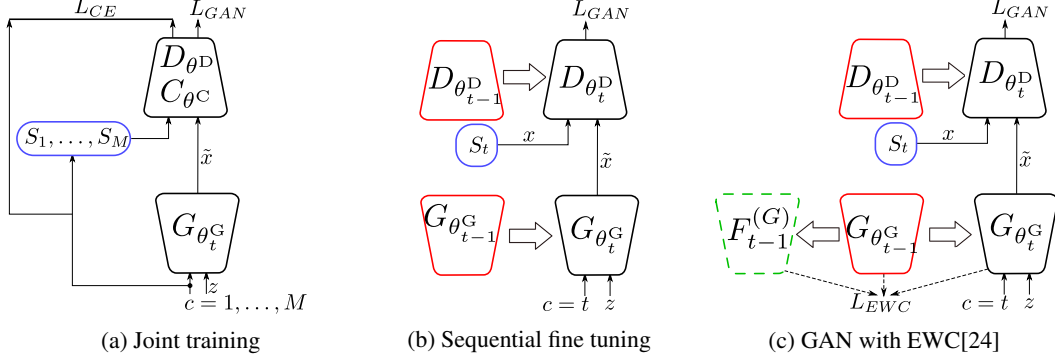

|  (a) Joint training | (b) Sequential fine tuning | (c) GAN with EWC[24] |

Figure 1: Baseline architectures.

While previous works study forgetting in discriminative tasks, in this paper we focus on forgetting in generative models (GANs in particular) through the problem of generating images when categories are presented sequentially as disjoint tasks. The closest related work is [24], that adapts elastic weight consolidation (EWC) [4] to GANs. In contrast, our method relies on memory replay and we describe two approaches to prevent forgetting by joint retraining and by aligning replays. The former includes replayed samples in the training process, while the latter forces to synchronize the replays of the current generator with those generated by an auxiliary generator (a snapshot taken before starting to learn the new task). An advantage of studying forgetting in image generation is that the dynamics of forgetting and consolidation can be observed visually through the generated images themselves.

## 2 Sequential learning in GANs

### 2.1 Joint learning

We first introduce our conditional GAN framework in the non-sequential setting, where all categories are learned jointly. In particular, this first baseline is based on the AC-GAN framework [20] combined with the WGAN-GP loss for robust training [7]. Using category labels as conditions, the task is to learn from a training set $S = \{S_1, \ldots, S_M\}$ to generate images given an image category $c$. Each set $S_c$ represents the training images for a particular category.

The framework consists of three components: generator, discriminator and classifier. The discriminator and classifier share all layers but the last ones (task-specific layers). The conditional generator is parametrized by $\theta^G$ and generates an image $\tilde{x} = G_{\theta^G}(z, c)$ given a latent vector $z$ and a category $c$. In our case the conditioning is implemented via conditional batch normalization [3], that dynamically switches between sets of batch normalization parameters depending on $c$. Note that, in contrast to unconditional GANs, the latent vector is completely agnostic to the category, and the same latent vector can be used to generate images of different categories just by using a different $c$.

Similarly, the discriminator (parametrized by $\theta^D$) tries to discern whether an input image $x$ is real (i.e. from the training set) or generated, while the generator tries to fool it by generating more realistic images. In addition, AC-GAN uses an auxiliary classifier $C$ with parameters $\theta^C$ to predict the label $\tilde{c} = C_{\theta^C}(x)$, and thus forcing the generator to generate images that can be classified in the same way as real images. This additional task improves the performance in the original task [20]. For convenience we represent all the parameters in the conditional GAN as $\theta = (\theta^G, \theta^D, \theta^C)$.

During training, the network is trained to solve both the adversarial game (using the WGAN with gradient penalty loss [7]) and the classification task by alternating the optimization of the generator, and the discriminator and classifier. The generator optimizes the following problem:

$$\min_{\theta^G} \left( L_{\text{GAN}}^{\text{G}}(\theta, S) + L_{\text{CLS}}^{\text{G}}(\theta, S) \right) \tag{1}$$

$$L_{\text{GAN}}^{\text{G}}(\theta, S) = -\mathbb{E}_{z \sim p_z, c \sim p_c} \left[ D_{\theta^D}(G_{\theta^G}(z, c)) \right] \tag{2}$$

$$L_{\text{CLS}}^{\text{G}}(\theta, S) = -\mathbb{E}_{z \sim p_z, c \sim p_c} \left[ y_c \log C_{\theta^C}(G_{\theta^G}(z, c)) \right] \tag{3}$$

where $L_{\text{GAN}}^{\text{G}}(\theta, S)$ and $\lambda_{\text{CLS}} L_{\text{CLS}}^{\text{G}}(\theta, S)$ are the corresponding GAN and cross-entropy loss for classification, respectively, $S$ is the training set, $p_c = \mathcal{U}\{1, M\}$, $p_z = \mathcal{N}(0, 1)$ are the sampling distributions (uniform and Gaussian, respectively), and $y_c$ is the one-hot encoding of $c$ for computing the cross-entropy. The GAN loss uses the WGAN formulation with gradient penalty. Similarly, the optimization problem in the discriminator and classifier is

$$\min_{\theta^{\text{D}}, \theta^{\text{C}}} \left( L_{\text{GAN}}^{\text{D}}(\theta, S) + \lambda_{\text{CLS}} L_{\text{CLS}}^{\text{D}}(\theta, S) \right) \tag{4}$$

$$L_{\text{GAN}}^{\text{D}}(\theta, S) = -\mathbb{E}_{(x,c)\sim S}\left[D_{\theta^{\text{D}}}(x)\right] + \mathbb{E}_{z\sim p_z, c\sim p_c}\left[D_{\theta^{\text{D}}}(G_{\theta^{\text{G}}}(z, c))\right] \tag{5}$$

$$+ \lambda_{\text{GP}} \mathbb{E}_{x\sim S, z\sim p_z, c\sim p_c, \epsilon\sim p_\epsilon} \left[ \left( \| \nabla D_{\theta^{\text{D}}}(\epsilon x + (1-\epsilon) G_{\theta^{\text{G}}}(z, c)) \|_2 - 1 \right)^2 \right]$$

$$L_{\text{CLS}}^{\text{D}}(\theta, S) = -\mathbb{E}_{(x,c)\sim S}\left[C_{\theta^{\text{C}}}(G_{\theta^{\text{G}}}(z, c))\right] \tag{6}$$

where $\epsilon$ are parameters of the gradient penalty term, sampled as $p_\epsilon = \mathcal{U}(0, 1)$. The last term of $L_{\text{GAN}}^{\text{D}}$ is the gradient penalty.

## 2.2 Sequential fine tuning

Now we modify the previous framework to address the sequential learning scenario. We define a sequence of tasks $\mathbf{T} = (1, \ldots, M)$, each of them corresponding to learning to generate images from a new training set $S_t$. For simplicity, we restrict each $S_t$ to contain only images from a particular category $c$, i.e. $t = c$.

The joint training problem can be adapted easily to the sequential learning scenario as

$$\min_{\theta_t^{\text{G}}} L_{\text{GAN}}^{\text{G}}(\theta_t, S_t) \tag{7}$$

$$\min_{\theta_t^{\text{D}}} L_{\text{GAN}}^{\text{D}}(\theta_t, S_t) \tag{8}$$

where $\theta_t = \left(\theta_t^G, \theta_t^D\right)$ are the parameters during task $t$, which are initialized as $\theta_t = \theta_{t-1}$, i.e. the current task $t$ is learned immediately after finishing the previous task $t - 1$. Note that there is no classifier in this case since there is only data of the current category.

Unfortunately, when the network learns to adjust its parameters to generate images of the new domain via gradient descent, that very drifting away from the original solution for the previous task will cause catastrophic forgetting [17]. This has also been observed in GANs [24, 27] (shown later in Figures 3, 5 and 7 in the experiments section).

## 2.3 Preventing forgetting with Elastic Weight Consolidation

Catastrophic forgetting can be alleviated using samples from previous tasks [22, 15] or different types of regularization that result in penalizing large changes in parameters or activations [4, 13]. In particular, the elastic weight consolidation (EWC) regularization [4] has been adapted to prevent forgetting in GANs [24] and included as an augmented objective when training the generator as

$$\min_{\theta_t^{\text{G}}} L_{\text{GAN}}^{\text{G}}(\theta_t, S_t) + \sum_i \frac{\lambda_{EWC}}{2} F_{t-1,i} \left( \theta_{t,i}^{\text{G}} - \theta_{t-1,i}^{\text{G}} \right)^2 \tag{9}$$

where $F_{t-1,i}$ is the Fisher information matrix that somewhat indicates how sensitive the parameter $\theta_{t,i}^{\text{G}}$ is to forgetting, and $\lambda_{EWC}$ is a hyperparameter. We will use this approach as a baseline.

## 3 Memory replay generative adversarial networks

Rather than regularizing the parameters to prevent forgetting, we propose that the generator has an active role by replaying memories of previous tasks (via generative sampling), and using them during the training of current task to prevent forgetting. Our framework is extended with a replay generator, and we describe two different methods to leverage memory replays.

This replay mechanism (also known as pseudorehearsal [23]) resembles the role of the hippocampus in replaying memories during memory consolidation [5], and has been used to prevent forgetting in classifiers [10, 25], but to our knowledge has not been used to prevent forgetting in image generation. Note also that image generation is a generative task and typically more complex than classification.

## 3.1 Joint retraining with replayed samples

Our first method to leverage memory replays creates an extended dataset $S'_t = S_c \bigcup_{c \in \{1,\dots,t-1\}} \tilde{S}_c$ that contains both real training data for the current tasks and memory replays from previous tasks. The replay set $\tilde{S}_c$ for a given category $c$ typically samples a fixed number for replays $\hat{x} = G_{\theta^G_{t-1}}(z, c)$.

Once the extended dataset is created, the network is trained using joint training (see Fig. 2a) as

$$\min_{\theta^G_t} \left( L^G_{GAN}(\theta_t, S'_t) + \lambda_{CLS} L^G_{CLS}(\theta_t, S'_t) \right) \tag{10}$$

$$\min_{\theta^D_t} \left( L^D_{GAN}(\theta_t, S'_t) + \lambda_{CLS} L^D_{CLS}(\theta_t, S'_t) \right) \tag{11}$$

This method could be related to the deep generative replay in [25], where the authors use an unconditional GAN and the category is predicted with a classifier. In contrast, we use a conditional GAN where the category is an input, allowing us finer control of the replay process, with more reliable sampling of $(x, c)$ pairs since we avoid potential classification errors and biased sampling towards recent categories.

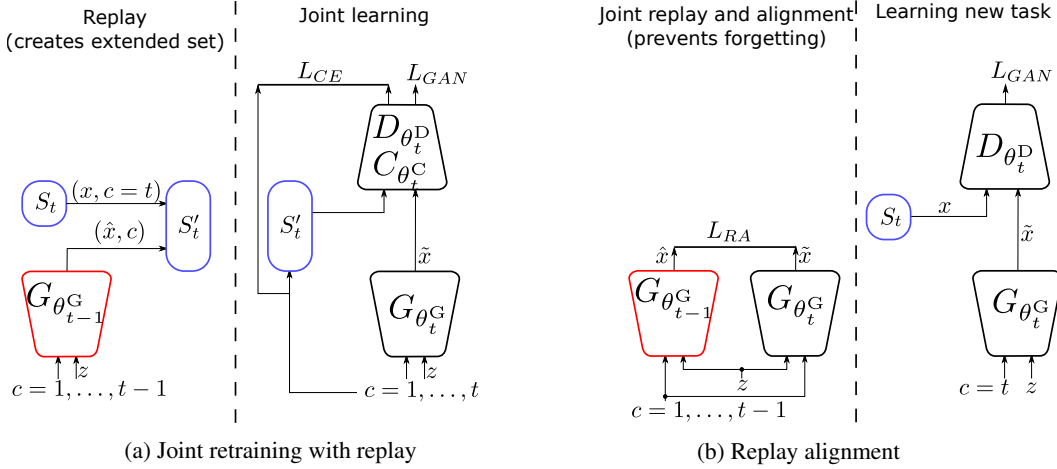

(a) Joint retraining with replay        (b) Replay alignment

Figure 2: Memory Replay GANs and mechanisms to prevent forgetting (for a given current task $t$).

## 3.2 Replay alignment

We can also take advantage of the fact that the current generator and the replay generator share the same architecture, inputs and outputs. Their condition spaces (i.e. categories), and, critically, their latent spaces (i.e. latent vector $z$) and parameter spaces are also initially aligned, since the current generator is initialized with the same parameters of the replay generator. Therefore, we can synchronize both the replay generator and current one to generate the same image by the same category $c$ and latent vector $z$ as inputs (see Fig. 2b). In these conditions, the generated images $\hat{x}$ and $x$ should also be aligned pixelwise, so we can include a suitable pixelwise loss to prevent forgetting (we use $L_2$ loss).

In contrast to the previous method, in this case the discriminator is only trained with images of the current task, and there is no classification task. The problem optimized by the generator includes a replay alignment loss

$$\min_{\theta^G_t} L^G_{GAN}(\theta_t, S_t) + \lambda_{RA} L_{RA}(\theta_t, S_t) \tag{12}$$

$$L_{RA}(\theta_t, S_t) = \mathbb{E}_{x \sim S, z \sim p_z, c \sim \mathcal{U}\{1, t-1\}} \left[ \left\| G_{\theta^G_t}(z, c) - G_{\theta^G_{t-1}}(z, c) \right\|^2 \right] \tag{13}$$

Note that in this case both generators engage in memory replay for all previous tasks. The corresponding problem in the discriminator is simply $\min_{\theta^D_t} L^D_{GAN}(\theta_t, S_t)$.

Our approach can be seen as *aligned distillation*, where distillation requires spatially aligned data. Note that in that way it could be related to the *learning without forgetting* approach [13] to prevent forgetting. However, we want to emphasize several subtle yet important differences:

**Different tasks and data** Our task is image generation where outputs have a spatial structure (i.e. images), while in [13] the task is classification and the output is a vector of category probabilities.

**Spatial alignment** Image generation is a one-to-many task with many latent factors of variability (e.g. pose, location, color) that can result in completely different images yet sharing the same input category. The latent vector $z$ somewhat captures those factors and allows an unique solution for a given $(z, c)$. However, pixelwise comparison of the generated images requires that not only the input but also the output representations are aligned, which is ensured in our case since at the beginning of training both have the same parameters. Therefore we can use a pixelwise loss.

**Task-agnostic inputs and seen categories** In [13], images of the current classification task are used as inputs to extract output features for distillation. Note that this implicitly involves a domain shift, since a particular input image is always linked to an unseen category (by definition, in the sequential learning problem the network cannot be presented with images of previous tasks), and therefore the outputs for the old task suffer from domain shift. In contrast, our approach does not suffer from that problem since the inputs are not real data, but a category-agnostic latent vector $z$ and a category label $c$. In addition, we only replay seen categories for both generators, i.e. 1 to $t - 1$.

## 4   Experimental results

We evaluated the proposed approaches in different datasets with different level of complexity. The architecture and settings are set accordingly. We use the Tensorflow [1] framework with Adam optimizer [11], learning rate 1e-4, batch size 64 and fixed parameters for all experiments: $\lambda_{EWC}$ = 1e9, $\lambda_{RA}$ = 1e-3 and $\lambda_{CLS}$ = 1 except for $\lambda_{RA}$ = 1e-2 on SVHN dataset.

### 4.1   Digit generation

We first consider the digit generation problem in two standard digit datasets. Learning to generate a digit category is considered as a separate task. MNIST [12] consists of images of handwritten digits which are resized $32 \times 32$ pixels in our experiment. SVHN [19] contains cropped digits of house numbers from real-world street images. The generation task is more challenging since SVHN contains much more variability than MNIST, with diverse backgrounds, variable illumination, font types, rotated digits, etc.

The architecture used in the experiments is based on the combination of AC-GAN and Wasserstein loss described in Section 2.1. We evaluated the two variants of the proposed memory replay GANs: joint training with replay (MeRGAN-JTR) and replay alignment (MeRGAN-RA). As upper and lower bounds we also evaluated joint training (JT) with all data (i.e. non-sequential) and sequential fine tuning (SFT). We also implemented two additional methods based on related works: the adaptation of EWC to conditional GANs proposed by [24], and the deep generative replay (DGR) module of [25], implemented as an unconditional GAN followed by a classifier to predict the label. For experiments with memory replay we generate one batch of replayed samples (including all tasks) for every batch of real data. We use a three layer DCGAN [21] for both datasets. In order to compare the methods in a more challenging setting, we keep the capacity of the network relatively limited for SVHN.

Figure 3 compares the images generated by the different methods after sequentially training the ten tasks. Since DGR is unconditional, the category for visualization is the one predicted by its classifier. We observe that SFT completely forgets previous tasks in both datasets, while the other methods show different degrees of forgetting. The four methods are able to generate MNIST digits properly, although both MeRGANs show sharper ones. In the more challenging setting of SVHN (note that the JT baseline also struggles to generate realistic images), the digits generated by EWC are hardly recognizable, while DGR is more unpredictable, sometimes generates good images but often generating images with ambiguous digits. Those generated by MeRGANs are in general clear and more recognizable, but still showing some degradation due to the limited capacity of the network.

We also trained a classifier with real data, using classification accuracy as a proxy to evaluate forgetting. The rationale behind is that in general bad quality images will confuse the classifier and

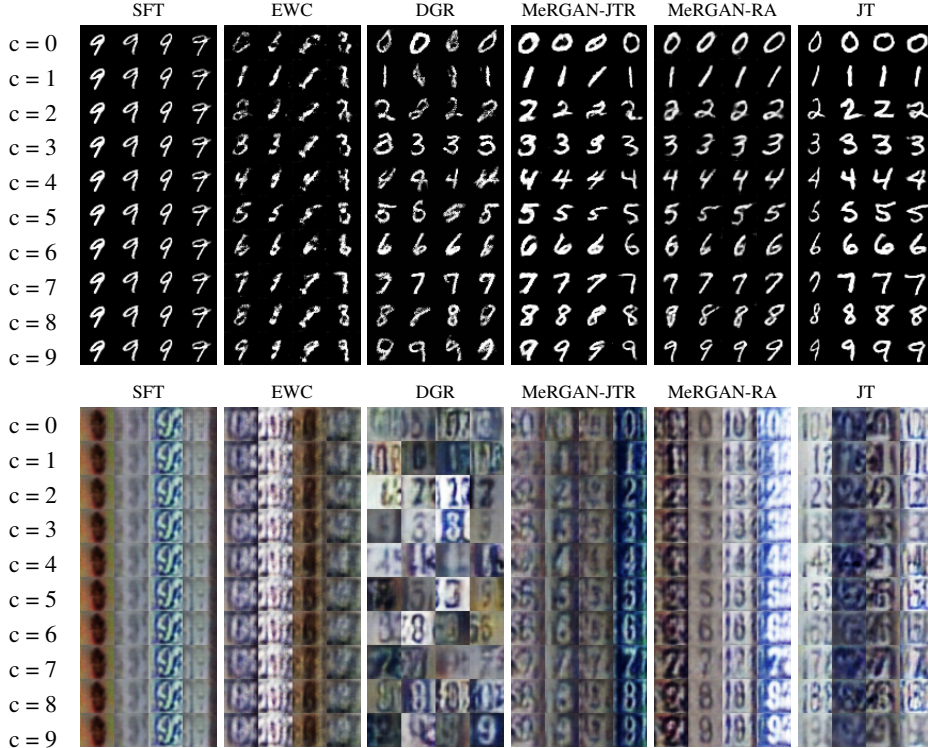

Figure 3: Images generated for MNIST and SVHN after learning the ten tasks. Rows are different conditions (i.e. categories), and columns are different latent vectors.

Table 1: Average classification accuracy (%) in digit generation (ten sequential tasks).

| | 5 tasks (0-4) | | | | | | 10 tasks (0-9) | | | | | |
| | Baselines | | Others | | MeRGAN | | Baselines | | Others | | MeRGAN | |
| | JT | SFT | EWC[24] | DGR[25] | JTR | RA | JT | SFT | EWC[24] | DGR[25] | JTR | RA |
|---|---|---|---|---|---|---|---|---|---|---|---|---|
| MNIST | 97.66 | 19.87 | 70.62 | 90.39 | **97.93** | **98.19** | 96.92 | 10.06 | 77.03 | 85.40 | **97.00** | **97.01** |
| SVHN | 85.30 | 19.35 | 39.84 | 61.29 | **80.90** | **76.05** | 84.82 | 10.10 | 33.02 | 47.28 | **66.50** | **66.78** |

result in lower classification rates. Table 1 shows the classification accuracy after the first five tasks (digits 0 to 4) and after the ten tasks. SFT forgets previous tasks so the accuracy is very low. As expected, EWC performs worse than DGR since it does not leverage replays, however it significantly mitigates the phenomenon of catastrophic forgetting by increasing the accuracy from 19.87 to 70.62 on MNIST, and from 19.35 to 39.84 on SVHN compared to SFT in the case of 5 tasks. The same conclusion can be drawn in the case of 10 tasks. By using the memory replay mechanism, MeRGANs obtain significant improvement compared to the baselines and the others related methods. Especially, our approach performs about 8% better on MNIST and about 21% better on SVHN compared to the strong baseline DGR in the case of 5 tasks. Note that our approach achieves about 12% gain in the case of 10 tasks, which shows that our approach is much more stable with increasing number of tasks. In the more challenging SVHN dataset, all methods decrease in terms of accuracy, however MerGAN are able to mitigate forgetting and obtain comparable results to JT.

Another interesting way to compare the different methods is through t-SNE visualizations. We use a classifier trained with real digits to extract embeddings of the methods to compare. Fig. 4a shows real 0s from MNIST and generated 0s from the different methods after training 10 tasks (i.e. the first task, and therefore the most difficult to remember). In contrast to SFT and EWC, the distributions of 0s generated by MeRGANs greatly overlap with the distribution of real 0s (in red) and no isolated clusters of real samples are observed, which suggests that MeRGANs prevent forgetting better while keeping diversity (at least in the t-SNE visualizations). Fig. 4b shows the t-SNE visualizations of real and 0s generated after learning 0,1,3 and 9, with similar conclusions.

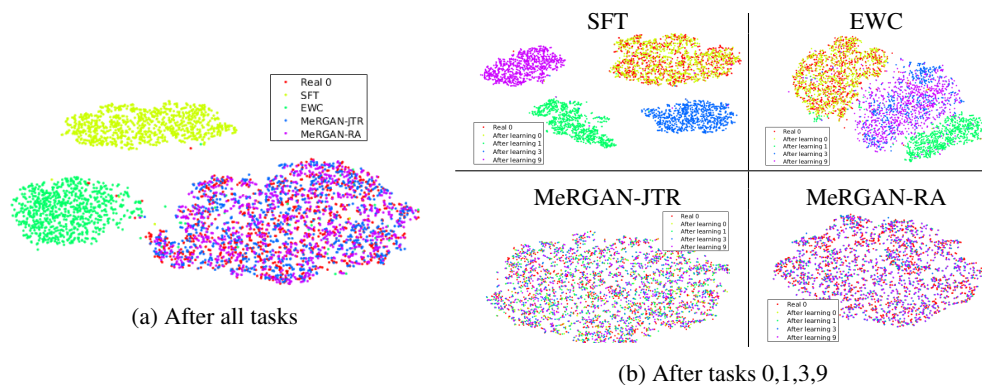

(a) After all tasks

(b) After tasks 0,1,3,9

Figure 4: t-SNE visualization of generated 0s. Real 0s correspond to red dots. Please view in electronic format with zooming.

Table 2: FID and average classification accuracy (%) on LSUN after the 4th task

|            | SFT    | EWC    | DGR   | MeRGAN-JTR | MeRGAN-RA |
|------------|--------|--------|-------|------------|-----------|
| Acc.(%)    | 15.02  | 14.28  | 15.40 | 79.19      | **81.03** |
| Rev acc.(%)| 28.0   | 63.35  | 26.17 | 70.00      | **83.62** |
| FID        | 110.12 | 178.05 | 93.70 | 49.69      | **37.73** |

## 4.2 Scene generation

We also evaluated MeRGANs in a more challenging domain and on higher resolution images ($64 \times 64$ pixels) using four scene categories of the LSUN dataset [28]. The experiment consists of a sequence of tasks, each one involving learning the generative distribution of a new category. The sequence of categories is *bedroom*, *kitchen*, *church (outdoors)* and *tower*, in this order. This sequence allows us to have two indoor and outdoor categories, and transitions between relatively similar categories (*bedroom* to *kitchen* and *church* to *tower*) and also a transition between very different categories (*kitchen* to *church*). Each category is represented by a set of 100000 training images, and the network is trained during 20000 iterations for every task. The architectures are based on [7] with 18-layer ResNet generator and discriminator, and for every batch of training data for the new category we generate a batch of replayed images per category.

Figure 5 shows examples of generated images. Each block column corresponds to a different method, and inside, each row shows images generated for a particular condition (i.e. category) and each column corresponds to images generated after learning a particular task, using the same latent vector. Note that we excluded DGR since the generation is not conditioned on the category. We can observe that SFT completely forgets the previous task, and essentially ignores the category condition. EWC generates images that have characteristics of both new and previous tasks (e.g. bluish outdoor colors, indoor shapes), being unable to neither successfully learn new tasks nor remember previous ones. In contrast both variants of MeRGAN are able to generate competitive images of new categories while still remembering to generate images of previous categories.

In addition to classification accuracy (using a VGG [26] trained over the ten categories in LSUN), for this dataset we add two additional measurements. The first one is reverse accuracy measured by a classifier trained with generated data and evaluated with real data. The second one is the Frechet Inception Distance (FID), which is widely used to evaluate the images generated by GANs. Note that FID is sensitive to both quality and diversity[8]. Table 2 shows these metrics after the four tasks are learned. MeRGANs perform better in this more complex and challenging setting, where EWC and DGR are severely degraded.

Figure 6 shows the evolution of these metrics during the whole training process, including transitions to new tasks (the curves have been smoothed for easier visualization). We can observe not only that sequential fine tuning forgets the task completely, but also that it happens early during the first few iterations. This also allows the network to exploit its full capacity to focus on the new task and learn it quickly. MeRGANs experience forgetting during the initial iterations of a new task but then tend to

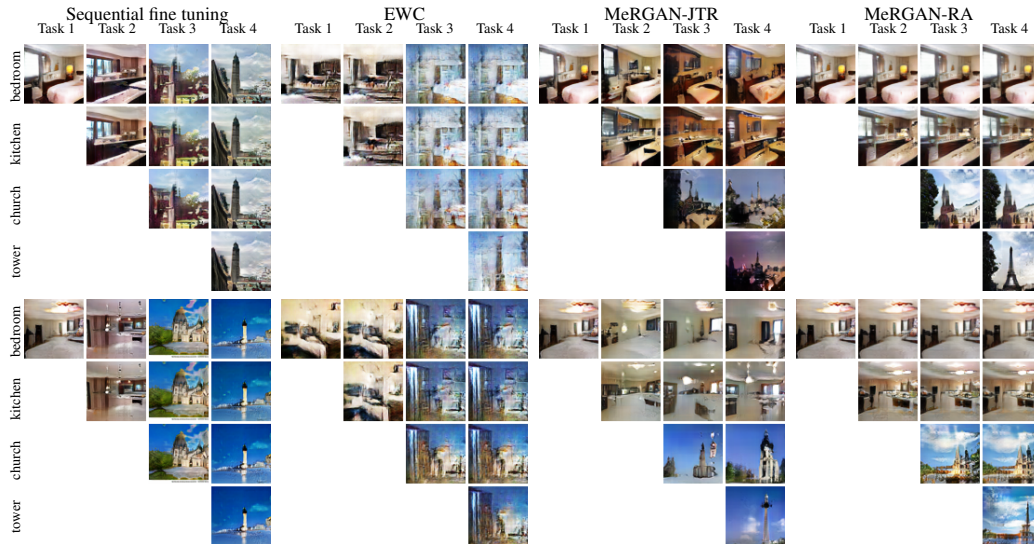

Figure 5: Images generated after sequentially learning each task (column within each block) for different methods (block column), two different latent vectors $z$ (block row) and different conditions $c$ (row within each block). The network learned after the first task is the same in all methods. Note that fine tuning forgets previous tasks completely, while the proposed methods still remember them.

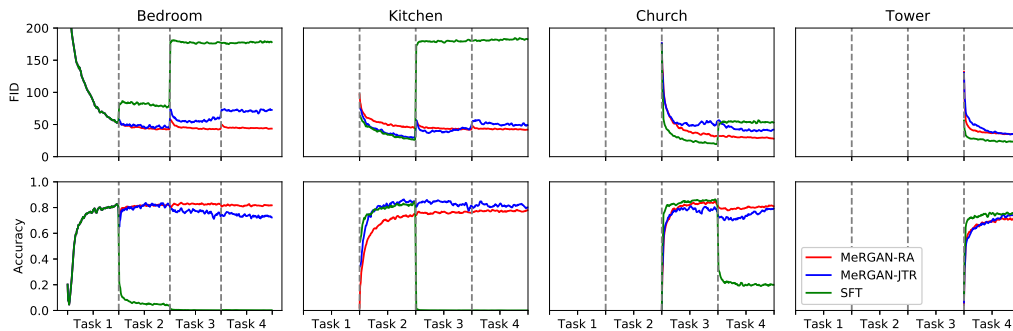

Figure 6: Evolution of FID and classification accuracy (%). Best viewed in color.

recover during the training process. In this experiment MeRGAN-RA seems to be more stable and slightly more effective than MeRGAN-JTR.

Figure 6 provides useful insight about the dynamics of learning and forgetting in sequential learning. The evolution of generated images also provides complementary insight, as in the *bedroom* images shown in Figure 7, where we pay special attention to the first iterations. The transition between task 2 to 3 (i.e. *kitchen* to *church*) is particularly revealing, since this new task requires the network to learn to generate many completely new visual patterns found in outdoor scenes. The most clear example is the need to develop filters that can generate the blue sky regions, that are not found in the previous indoor categories seen during task 1 and 2. Since the network is not equipped with knowledge to generate the blue sky, the new task has to reuse and adapt previous one, interfering with previous tasks and causing forgetting. This interference can be observed clearly in the first iterations of task 3 where the walls of bedroom (and kitchen) images turn blue (also related with the peak in forgetting observed at the same iterations in Figure 6). MeRGANs provide mechanisms that penalize forgetting, forcing the network to develop separate filters for the different patterns (e.g. separated filters for wall and sky). MeRGAN-JTR seems to effectively decouple both patterns, since we do not observe the same "blue walls" interference during task 4. Interestingly, the same interference seems to be milder in MeRGAN-RA, but recurrent, since it also appears again during task 4. Nevertheless, the interference is still temporary and disappears after a few iterations more.

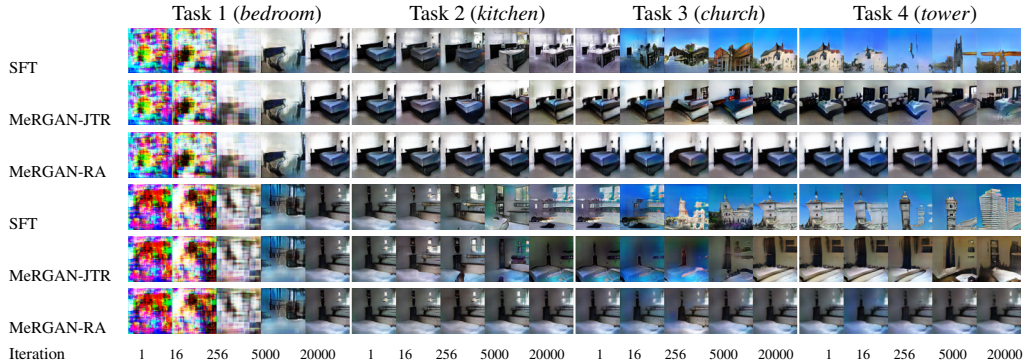

Figure 7: Evolution of the generated images (category *bedroom* and two different values of $z$) during the sequential learning process (rows). Sequential fine tuning forgets the previous task after just a few iterations (iterations within each task are sampled in a logarithmic fashion). Note that fine tuning forgets previous tasks completely, while the MeRGANs still remember them.

Another interesting observation from Figures 5 and 7 is that MeRGAN-RA remembers the *same* bedroom (e.g. same point of view, colors, objects), which is related to the replay alignment mechanism that enforces remembering the instance. On the other hand, MeRGAN-JTR remembers bedrooms *in general* as the generated image still resembles a bedroom but not exactly the same one as in previous steps. This can be explained by the fact that the classifier and the joint training mechanism enforce the not-forgetting constraint at the category level.

## Conclusions

We have studied the problem of sequential learning in the context of image generation with GANs, where the main challenge is to effectively address catastrophic forgetting. MeRGANs incorporate memory replay as the main mechanism to prevent forgetting, which is then enforced through either joint training or replay alignment. Our results show their effectiveness in retaining the ability to generate competitive images of previous tasks even after learning several new ones. In addition to the application in pure image generation, we believe MeRGANs and generative models robust to forgetting in general, could have important application in many other tasks. We also showed that image generation provides an interesting way to visualize the interference between tasks and potential forgetting by directly observing generated images.

## Acknowledgements

C. Wu, X. Liu, and Y. Wang, acknowledge the Chinese Scholarship Council (CSC) grant No.201709110103, No.201506290018 and No.201507040048. Luis Herranz acknowledges the European Union research and innovation program under the Marie Skłodowska-Curie grant agreement No. 6655919. This work was supported by TIN2016-79717-R, and the CHISTERA project M2CR (PCIN-2015-251) of the Spanish Ministry, the ACCIO agency and CERCA Programme / Generalitat de Catalunya, and the EU Project CybSpeed MSCA-RISE-2017-777720. We also acknowledge the generous GPU support from NVIDIA.

## Footnotes

[1] The code is available at `https://github.com/WuChenshen/MeRGAN`

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
