[Reviews · NeurIPS 2018]

Reviewer 1



This paper studies catastrophic forgetting problem of GAN generator under the continual learning setting, the old training data is stored in the generator of GAN, it is retrieved and replayed to the generator when training new categories. Quality and Clarity, Section 2 reviews the previous work in this problem, which is very clearly organized and easy to follow. The idea is straight forward and easy to understand. I like the fact that rehearsal of previous data can be converted into a regression problem (replay alignment). Originality and Significance: This work is original in the sense that no previous work has tackled the this specific problem using replay generated from GAN. However, the novelty is limited in that using GAN generator to store rehearsal examples is not new. Having a category input (input to the generator is both c and z) makes this a easier problem because it is aware of the category during training and generation. One may as well just use an individual network for each category if the category is known. In many realistic cases the task boundary is unknown.

Reviewer 2



This paper studies continual learning with generative adversarial networks. Specifically, two methods are proposed: 1) joint retraining with replayed samples 2) replay alignment. This paper addressed relatively important problem and suggest a reasonable baseline. I have concerns with the novelty of the proposed method because "joint retraining" and replay alignments are already employed in two papers referenced in each sections and application of the method in this problem seems straightforward. However, I believe applying these method to new important domain is valuable enough to be published if the experiments are solid. My another concern is the evaluation setting, where diversity of generated sampled in each class is not highlighted. - In the visualisations, only a single sample for each class is visualised. This visualisation make it hard to evaluate the catastrophic forgetting in terms of the sample diversity. I am not sure the proposed method keeps the model generate diverse sampled of previously learned task. - In the evaluation based on the classification accuracy, I believe sampled images should be used as a training examples, instead of the test examples. This is to evaluate the sample diversity. If generated samples for each class are identical, it is easy to achieve high accuracy if the samples are used as a test set, but it would be difficult to achieve high accuracy if the samples are used as a training set. Therefore I suggest to train classifier on samples and evaluate on oracle dataset (with true images). [Update after author response] My primary concern about this paper was experimental setting and diversity within class. These concerns are effectively addressed in the author response and I am convinced with the results. I highly recommend authors to include discussion related to the updated results in the main paper if the paper is accepted.

Reviewer 3



Update following the author rebuttal: I would like to thank the authors for their thoughtful rebuttal. I feel like they appropriately addressed the main points I raised, namely the incomplete evaluation and the choice of GANs over other generative model families, and I'm inclined to recommend the paper's acceptance. I updated my review score accordingly. ----- The paper presents a conditional GAN framework to combat catastrophic forgetting when learning new image categories in a sequential fashion. The paper is well-written and its exposition of the problem, proposed solution, and related work is clear. Starting from the AC-GAN conditional generative modeling formulation, the authors introduce the notion of a sequence of tasks by modeling image classes (for MNIST, SVHN, and LSUN) in sequence, where the model for each class in the sequence is initialized with the model parameters for the previous class in the sequence. They note that in this setup the AC-GAN’s classifier cannot be used, as for any given task there is only one class being trained on. The idea put forward is to use the generator as a memory replay mechanism to prevent forgetting when training for the current task. Two approaches are considered: - A joint replay approach (MerGAN-JTR), where upon completing a training task the model is sampled from, and the samples for previous tasks are combined with the dataset for the current task to form a hybrid real/synthetic class-conditional dataset. In this setup the AC-GAN formulation is used. - A replay alignment approach (MeRGAN-RA), where class-conditional samples for previous tasks are drawn continuously throughout training using the current value of the generator’s class-independent parameters and are compared with the class-conditional samples of the hybrid dataset using an L2 pixelwise loss (using the same latent samples z for both). In this setup the AC-GAN’s classifier is not used. The two approaches, along with several baselines -- naive sequential fine-tuning (SFT), elastic weight consolidation (EWC), deep generative replay (DGR), and joint training (JT) -- are trained on sequences of classes from MNIST, SVHN, and LSUN and evaluated qualitatively and quantitatively. For MNIST and SVHN, qualitative evaluation consists of examining class-conditional model samples after completing the last training task in the sequence and shows that both proposed approaches perform better than the competing EWC and DGR approaches (especially for SVHN). For LSUN, qualitative evaluation compares both proposed approaches with the naive SFT baseline and shows samples drawn after training on each task in the sequence. I would have liked to see a comparison with EWC and DGR here as well; can the authors tell how they compare? In terms of quantitative evaluation, the accuracy of an external classifier on class-conditional samples is presented for MNIST, SVHN, and LSUN, showing that the proposed approaches outperform the naive SFT baseline as well as the competing approaches. Once again, I would have liked to see a comparison with EWC and DGR for LSUN. I find it interesting that for MNIST the proposed approaches do better than the upper-bound JT baseline; do the authors have an idea why? I would like to see confidence intervals on the accuracies, as it may help decide whether the improved performance over JT is statistically significant or not. Models trained on LSUN are also evaluated in terms of Fréchet Inception Distance, showing again that the two proposed approaches perform much better than the naive SFT baseline. One criticism I would have to voice about the evaluation procedure is that the sequence of class-conditional tasks setup is a little contrived. I agree with the premise that in real-world scenarios the full training data is often not immediately available, but at the same time the sort of input distribution drifts we are likely to encounter is probably smoother than the discrete shifts in input distribution corresponding to switching the object class (an example would be the day/night cycle a camera pointed at a highway experiences). Moreover, information about the input distribution -- be it a discrete class label or some continuous attribute representation -- probably won’t be available in real-world scenarios. Having said that, I am aware this is an issue not unique to the proposed method, and I am okay with the use of the proposed evaluation setup given that most previous work on the topic also uses a similar evaluation setup -- and also given that the search for a better setup is still an open question to me and I don’t have a better idea to propose. Another criticism I have is that the choice of GANs as the training framework makes it unnecessarily difficult to compare and evaluate different methods: the right way of evaluating GANs is still very much an open problem, which means that any choice of GAN evaluation metric comes with its own set of caveats. For instance, one problem with the classifier accuracy metric is that the model could produce class-conditional samples achieving a high accuracy by collapsing the generator to only a few exemplars per class. Given that the proposed method appears to be applicable to any class-conditional generative model, I think the choice of an explicit likelihood model family along with a log-likelihood metric would have been easier to defend from an evaluation point of view. In summary, the paper presents a clear exposition of an idea to solve a very relevant research problem, namely catastrophic forgetting in generative models. I am a bit less enthusiastic about the empirical evaluation, mostly because I feel it is incomplete (as I said, I would like to see comparisons with competing methods for LSUN), and because the choice of model family and evaluation metric makes it harder to compare and evaluate competing methods.